# Distinctive whole-brain cell types predict tissue damage patterns in thirteen neurodegenerative conditions

Veronika Pak[1,2,3], Quadri Adewale[1,2,3], Danilo Bzdok[2,4,5,6], Mahsa Dadar[7], Yashar Zeighami[7], Yasser Iturria-Medina[1,2,3,4,8]*

[1]Department of Neurology and Neurosurgery, McGill University, Montreal, Canada; [2]McConnell Brain Imaging Centre, Montreal Neurological Institute, Montreal, Canada; [3]Ludmer Centre for Neuroinformatics & Mental Health, Montreal, Canada; [4]Department of Biomedical Engineering, McGill University, Montreal, Canada; [5]School of Computer Science, McGill University, Montreal, Canada; [6]Mila – Quebec Artificial Intelligence Institute, Montreal, Canada; [7]The Douglas Research Center, Montreal, Canada; [8]McGill Centre for Studies in Aging, Montreal, Canada

*For correspondence:
lturria.medina@gmail.com

Competing interest: The authors declare that no competing interests exist.

**Abstract** For over a century, brain research narrative has mainly centered on neuron cells. Accordingly, most neurodegenerative studies focus on neuronal dysfunction and their selective vulnerability, while we lack comprehensive analyses of other major cell types' contribution. By unifying spatial gene expression, structural MRI, and cell deconvolution, here we describe how the human brain distribution of canonical cell types extensively predicts tissue damage in 13 neuro-degenerative conditions, including early- and late-onset Alzheimer's disease, Parkinson's disease, dementia with Lewy bodies, amyotrophic lateral sclerosis, mutations in presenilin-1, and 3 clinical variants of frontotemporal lobar degeneration (behavioral variant, semantic and non-fluent primary progressive aphasia) along with associated three-repeat and four-repeat tauopathies and TDP43 proteinopathies types A and C. We reconstructed comprehensive whole-brain reference maps of cellular abundance for six major cell types and identified characteristic axes of spatial overlapping with atrophy. Our results support the strong mediating role of non-neuronal cells, primarily microglia and astrocytes, in spatial vulnerability to tissue loss in neurodegeneration, with distinct and shared across-disorder pathomechanisms. These observations provide critical insights into the multicel-lular pathophysiology underlying spatiotemporal advance in neurodegeneration. Notably, they also emphasize the need to exceed the current neuro-centric view of brain diseases, supporting the imperative for cell-specific therapeutic targets in neurodegeneration.

## eLife assessment

Pak et al. examined the relationship between the most common spatial patterns of neurodegeneration and transcriptional markers of the density of different cell types in the cerebral cortex. This **valuable** study uses innovative methods to provide **convincing** evidence that patterns of gray matter loss in various forms of dementia are correlated with the anatomical distribution of non-neuronal cell types.

## Introduction

Neurodegenerative diseases are characterized by substantial neuronal loss in both the central and peripheral nervous systems (*Gorman, 2008*). In dementia-related conditions like Alzheimer's disease

(AD), frontotemporal dementia (FTD), and dementia with Lewy bodies (DLB), neurodegeneration can lead to progressive damage in brain regions related with memory, behavior, and cognition (*Duong et al., 2017*). Other diseases are thought to primarily affect the locomotor system, including motor neurons in amyotrophic lateral sclerosis (ALS) and nigrostriatal dopaminergic circuitry in Parkinson's disease (PD) (*Bosco et al., 2011*). Although each disorder has its own distinct etiology, progression, affected brain areas, and clinical manifestations, recent studies support that most of them share same molecular and cellular mechanisms (*Wingo et al., 2022*; *Huseby et al., 2023*; *Arneson et al., 2018*; *Zeighami et al., 2023*).

While research has been mainly focused on neuronal dysfunction, other brain cells such as astrocytes, microglia, oligodendrocytes, as well as cells of the vascular and peripheral immune systems, are gaining more recognition for their contribution to disease pathology (*Bordone and Barbosa-Morais, 2020*; *Lee et al., 2020*; *Reynolds et al., 2019*). Depending on the disease stage, non-neuronal cells in the brain can play a dual role, with their complex response having both protective and detrimental effects on neuronal health and survival (*Geloso et al., 2017*; *Jiwaji et al., 2022*). For instance, such glial cells as astrocytes and microglia are involved in neuronal support, maintenance of extracellular homeostasis, and immune regulation in response to injury (*Garland et al., 2022*; *Kwon and Koh, 2020*). Initially, these cells respond to injury by releasing neuroprotective neurotrophic factors and antioxidants (*Garland et al., 2022*; *Kwon and Koh, 2020*). However, under certain conditions, prolonged microglial activation can induce reactive astrocytes and together they release neurotoxic pro-inflammatory cytokines and chemokines, which in turn can lead to metabolic stress and foster the accumulation of amyloid-β and tau plaques in AD, ultimately contributing to heightened neuronal death (*Kempuraj et al., 2016*; *Yun et al., 2018*; *Liddelow et al., 2017*). Growing evidence suggests that immune and other cell type-mediated events are a driving force behind the wide range of neurodegenerative conditions (*Kempuraj et al., 2016*; *Maccioni et al., 2009*; *Zang et al., 2022*; *Castellani et al., 2023*; *Balusu et al., 2023*). Yet, the exact bases behind how these processes contribute to selective neuronal loss across brain regions remain unclear.

Recent studies have suggested that brain spatial patterns in gene expression are associated with regional vulnerability to some neurodegenerative disorders and their corresponding tissue atrophy distributions (*Vidal-Pineiro et al., 2020*; *Roshchupkin et al., 2016*; *Zheng et al., 2019*; *Altmann et al., 2020*; *Kerrebijn et al., 2023*). A comparison of transcriptomic patterns in middle temporal gyrus across various brain diseases showed cell type expression signature unique for neurodegenerative diseases (*Zeighami et al., 2023*). Although single-cell transcriptomics and multiomics analyses have advanced our knowledge of cell type compositions associated with pathology in neurodegeneration (*Cuevas-Diaz Duran et al., 2022*; *Luquez et al., 2022*; *Riley et al., 2014*), these are invariably restricted to a few isolated brain regions, usually needing to be preselected at hand for each specific disease. Due to the invasive nature of tissue acquisition/mapping and further technical limitations for covering extended areas (*Arnatkeviciute et al., 2022*), no whole-brain maps for the abundance of cell populations in humans are currently available, constraining the analysis of large-scale cellular vulnerabilities in neurological diseases. Accordingly, how spatial cell type distributions relate to stereotypic regional damages in different neurodegenerative conditions remains largely unclear (*Mrdjen et al., 2019*).

Here, we extend previous analyses of cellular-based spatiotemporal vulnerability in neurodegeneration in three fundamental ways. First, we use transcriptomics, structural MRI, and advanced cell deconvolution to construct whole-brain reference maps of cellular abundance in healthy humans for six canonical cell types: neurons, astrocytes, oligodendrocytes, microglia, endothelial cells, and oligodendrocyte precursors. Second, we describe the spatial associations of each healthy level of reference canonical cell types with atrophy in 13 low-to-high prevalent neurodegenerative conditions, including early- and late-onset AD, genetic mutations in presenilin-1 (PS1 or PSEN1), DLB, ALS, PD, and both clinical and pathological subtypes of frontotemporal lobar degeneration (FTLD). Third, we identify distinctive cell–cell and disorder–disorder axes of spatial susceptibility in neurodegeneration, obtaining new insights about across-disorder (dis)similarities in underlying pathological cellular systems. We confirm that non-neuronal cells express substantial vulnerability to tissue loss and spatial brain alterations in most studied neurodegenerative conditions, with distinct and shared across-cell and across-disorder mechanisms. This study aids in unraveling the commonalities across a myriad of dissimilar neurological conditions, while also revealing cell type-specific patterns conferring increased

vulnerability or resilience to each examined disorder. For further translation and validation of our findings, all resulting analytic tools and cell abundance maps are shared with the scientific and clinical communities.

## Results

### Multimodal data origin and unification approach

We obtained whole-brain voxel-wise atrophy maps for 13 neurodegenerative conditions, including early- and late-onset Alzheimer's disease (EOAD and LOAD, respectively), PD, ALS, DLB, mutations in presenilin-1 (PS1), clinical variants of FTD (the behavioral variant [bvFTD] and the non-fluent and semantic variants of primary progressive aphasia [nfvPPA and svPPA]), and FTLD-related pathologies such as FLTD-TDP (TAR DNA-binding protein) types A and C, three-repeat tauopathy, and four-repeat tauopathy (see 'Disease-specific atrophy maps'; *Harper et al., 2017*; *Dadar et al., 2020*; *Zeighami et al., 2015*; *Dadar and Metz, 2023*). We use the term FTD when addressing the clinical syndromes, and the term FTLD is employed when referencing histologically confirmed neurodegenerative pathologies (*Boeve et al., 2022*). Pathological diagnosis confirmation was performed for EOAD and LOAD, DLB, PS1, FTLD-TDP types A and C, three-repeat tauopathy, and four-repeat tauopathy (*Harper et al., 2017*), while PD, ALS, and variants of FTD were diagnosed based on clinical and/or neuroimaging criteria (*Parkinson Progression Marker Initiative, 2011*; *Kalra et al., 2020*; *Staffaroni et al., 2019*), with some ALS patients being histologically confirmed *postmortem* (*Kalra et al., 2020*). Changes in tissue density in the atrophy maps were previously measured by voxel- and deformation-based morphometry (VBM and DBM; see 'Disease-specific atrophy maps') applied to structural T1-weighted MR images, and expressed as a *t*-score per voxel (relatively low negative values indicate greater GM tissue loss/atrophy; *Aubert-Broche et al., 2013*; *Ashburner and Friston, 2000*). All maps are registered to the Montreal Neurological Institute (MNI) brain space (*Evans et al., 1994*). In addition, we obtained bulk transcriptomic data for the adult healthy human brains from the Allen Human Brain Atlas (AHBA) (*Shen et al., 2012*). This included high-resolution coverage of nearly the entire brain, measuring expression levels for over 20,000 genes from 3702 distinct tissue samples of six postmortem specimens, and detailed structural MRI data (see 'Mapping gene expression data'; *Shen et al., 2012*).

Using a previously validated approach to infer gene expression levels (in AHBA data) at nonsampled brain locations with Gaussian process regression (*Gryglewski et al., 2018*), mRNA expression levels were completed for all gray matter (GM) voxels in the standardized MNI brain space (*Evans et al., 1994*). Gaussian process regression allowed predicting gene expression values for unobserved regions based on the mRNA values of proximal regions. Next, at each GM location, densities for multiple canonical cell types were estimated using the Brain Cell type-Specific Gene Expression Analysis software (BRETIGEA) (*McKenzie et al., 2018a*). The deconvolution method (*McKenzie et al., 2018a*; *Chikina et al., 2015*; implemented in the BRETIGEA) accurately estimated cell proportions from bulk gene expression for six major cell types (*Figure 1B*): neurons, astrocytes, oligodendrocytes, microglia, endothelial cells, and oligodendrocyte precursor cells (OPCs). Overall, atrophy levels for 13 neurodegenerative conditions and proportion values for 6 major cell types from healthy brains were unified at matched and standardized locations (MNI space), covering the entire GM of the brain (see *Figure 1* for a schematic description).

We hypothesized (and tested in next subsections) that brain tissue damages in neurodegenerative conditions are associated with distinctive patterns of cells distributions, with alterations on major cell types playing a key role on the development of each disorder and representing a direct factor contributing to brain dysfunction.

### Uncovering spatial associations between cell type abundances and tissue damage in neurodegeneration

First, we investigated whether stereotypic brain atrophy patterns in neurodegenerative conditions show systematic associations with the spatial distribution of canonical cell type populations in healthy brains. For each condition and cell type pair, the nonlinear Spearman's correlation coefficient was calculated with paired atrophy–cell proportion values across 118 cortical and subcortical regions defined by the automated anatomical labeling (AAL) atlas (*Tzourio-Mazoyer et al., 2002*; *Supplementary*

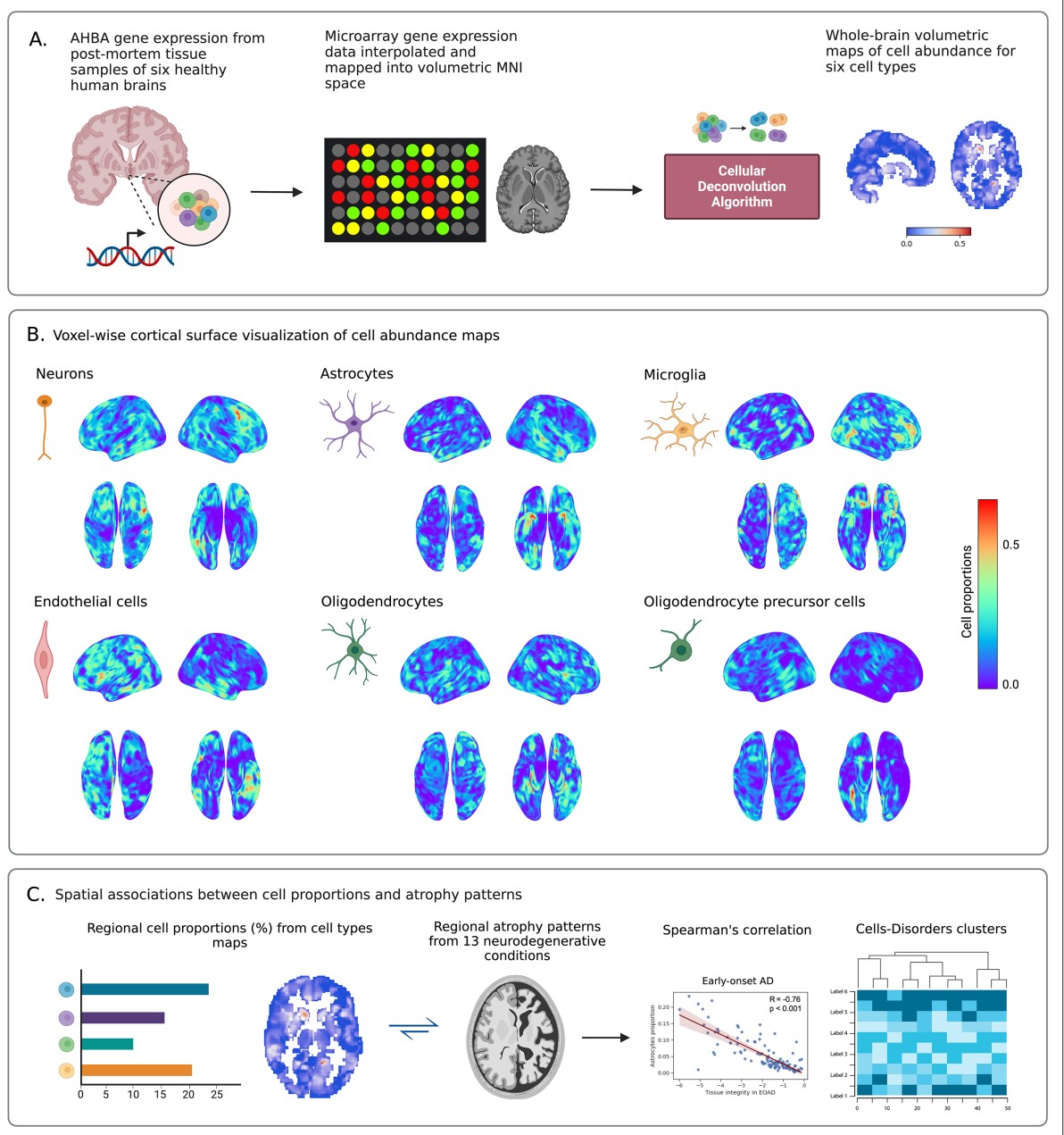

**Figure 1.** Schematic approach for whole-brain cell type proportions vulnerability analysis in neurodegeneration. (**A**) Microarray bulk gene expression levels in the Allen Human Brain Atlas (AHBA) were derived from 3072 distinct tissue samples of six postmortem healthy human brains. Missing gene expression data were then inferred for each unsampled gray matter voxel using Gaussian process regression. When combined with original AHBA data, they were mapped into volumetric Montreal Neurological Institute (MNI) space, resulting in the whole-brain transcriptional atlas. Deconvolution algorithm for bulk RNA expression levels was applied to the transcriptional atlas by using well-known cell type-specific gene markers to estimate cell type proportions. Comprehensive volumetric maps showing reconstructed distributions of six canonical cell types across all gray matter voxels in the brain were created (see 'Cell type proportion estimation'). (**B**) Voxel-wise surface visualization (lateral, dorsal, and ventral views) of cell abundance maps for neurons, astrocytes, microglia, endothelial cells, oligodendrocytes, and oligodendrocyte precursor cells (OPCs). At each voxel, red and blue colors indicate high and low proportion densities, respectively. (**C**) Associations between cell type proportions from each density map and atrophy values in 13 neurodegenerative conditions were analyzed in 118 gray matter regions predefined by the automated anatomical labeling (AAL) atlas.

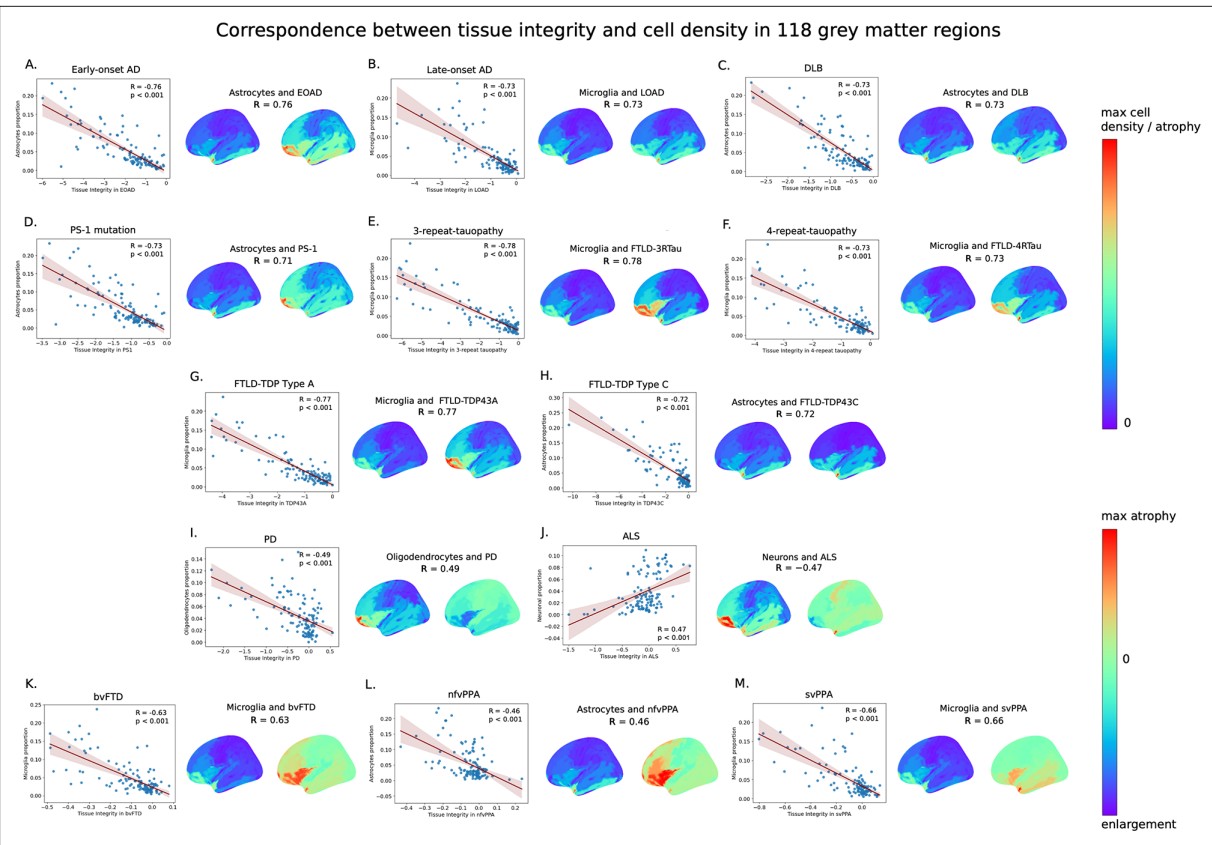

**Figure 2.** Spatial associations between tissue integrity and cell type proportions for 13 neurodegenerative conditions illustrated in the scatterplots and surface maps (left hemisphere; lateral view) of regional measures. (**A–M**) Strongest Spearman's correlations for early-onset Alzheimer's disease (EOAD), late-onset Alzheimer's disease (LOAD), dementia with Lewy bodies (DLB), presenilin-1 (PS1), FTLD-3Rtau, FTLD-4Rtau, FTLD-TDP43A, FTLD-TDP43C, Parkinson's disease (PD), amyotrophic lateral sclerosis (ALS), behavioral variant of frontotemporal dementia (bvFTD), non-fluent variant of primary progressive aphasia (nfvPPA), and semantic variant of primary progressive aphasia (svPPA), respectively. Atrophy and cell type density measures were averaged across 118 gray matter (GM) regions and projected to the cortical surface of the *fsaverage* template. Each dot in the scatterplots represents a GM region from the automated anatomical labeling (AAL) atlas (*Supplementary file 1*). Lower tissue integrity score in the scatterplots' x-axis indicates greater GM loss/atrophy. For a better visual comparison of patterns in atrophy and cell abundance, the atrophy scale was reversed, with higher *t*-statistic values indicating greater atrophy in the surface plots. Thus, the first color bar ranging from 0 is universal for all cell maps and pathologically confirmed dementia conditions (**A–H**). The second color bar captures the tissue enlargement in PD, ALS, and variants of FTD (**I–M**). Notice how astrocyte density significantly correlates with increase in tissue loss in EOAD, DLB, PS1, FTLD-TDP43C, and nfvPPA (**A, C, D, H, L**; p<0.001). Tissue loss was also associated with increase in microglial proportion in LOAD, FTLD-3Rtau, FTLD-4Rtau, FTLD-TDP43A, bvFTD, and svPPA (**B, E, F, G, K, M**; p<0.001). Increased oligodendrocytes associated with PD (**I**; p<0.001). Increase in neuronal proportion showed association with decrease in atrophy and tissue enrichment in ALS (**J**; p<0.001). All p-values were false discovery rate (FDR)-adjusted with the Benjamini–Hochberg procedure (p<0.05).

file 1). The results (*Figures 2A–M and 3A*) show clear associations for all the studied conditions, suggesting extensive cell type-related tissue damage vulnerability in neurodegenerative conditions. We confirmed that the observed relationships are independent of brain parcellation, obtaining equivalent results for a different brain parcellation (i.e., Desikan–Killiany–Tourville [DKT] atlas; *Desikan et al., 2006*; see *Figure 3—figure supplement 1*).

As shown in *Figures 2A–M and 3A*, astrocytes and microglia cell occurrences presented the strongest spatial associations with atrophy in most neurodegenerative conditions, particularly for EOAD, LOAD, DLB, PS1, FTLD-3RTau, FTLD-4Rtau, FTLD-TDP type A, FTLD-TDP type C, bvFTD, nfvPPA, and svPPA (all p<0.001, false discovery rate [FDR]-corrected). Astrocytes are involved in neuronal support, extracellular homeostasis, and inflammatory regulation in response to injury, and show high susceptibility to senescence and oxidative damage (*Li et al., 2021*; *González-Reyes et al., 2017*). Astrocytes also play an important role in the maintenance of the blood–brain barrier (BBB), which regulates the passage of molecules, ions, and cells between the blood and the brain (*Preininger and Kaufer, 2022*). A recent study suggested that reactive astrocytes may promote vascular inflammation

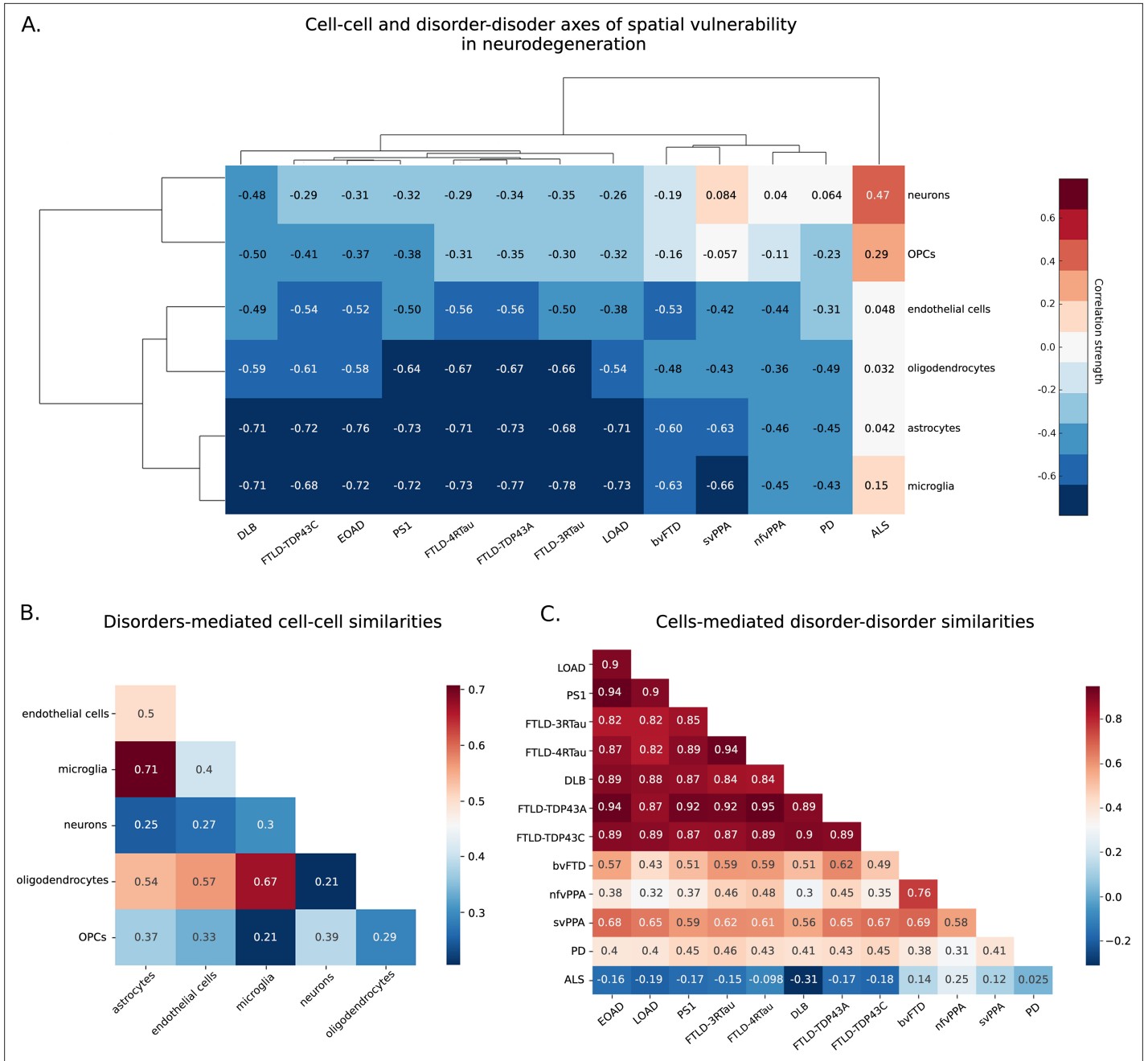

**Figure 3.** Cell and disorder similarities based on shared distributions. (**A**) Dendrogram and unsupervised hierarchical clustering heatmap of Spearman's correlations between cell type proportions and atrophy patterns across the 13 neurodegenerative conditions. (**B**) Cell–cell associations based on regional vulnerabilities to tissue loss across neurodegenerative conditions. (**C**) Disorder–disorder similarities across cell types. In (**A**), red color corresponds to strong positive correlations between cells and disorders, white to no correlation, and dark blue to strong negative correlations.

The online version of this article includes the following figure supplement(s) for figure 3:

**Figure supplement 1.** Spatial associations between tissue integrity and cell type proportions for 13 neurodegenerative conditions in the gray matter (GM) regions defined by the Desikan–Killiany–Tourville (DKT) parcellation.

in the BBB (***Kim et al., 2022***). Endothelial cells, which comprise the functional component of the BBB, also showed strong spatial associations with atrophy in almost all conditions (***Figure 3A***). Endothelial cells regulate cerebral blood flow and deliver oxygen and nutrients to the brain (***Pober and Sessa, 2007***). Disruption of the BBB may allow harmful substances to enter the brain, including inflammatory molecules and toxic-aggregated proteins, ultimately exacerbating neuronal damage (***Kalaria, 1997***;

*Salmina et al., 2010*). Reduction in cerebral blood flow and vascular dysregulation are the earliest and strongest pathological biomarkers of LOAD, PD, and other neurodegenerative disorders (*Iturria-Medina et al., 2016*; *Globus et al., 1985*; *Wolters et al., 2020*).

Similar to astrocytes in their role of supporting neurons, microglial cells are the resident macrophages of the central nervous system and key players in the pathology of neurodegenerative conditions, including AD, PD, FTD, and ALS (*Geloso et al., 2017*; *Guzman-Martinez et al., 2019*; *Malpetti et al., 2021*). Besides its many critical specializations, microglial activation in prolonged neuroinflammation is of particular relevance in neurodegeneration (*Geloso et al., 2017*; *Perea et al., 2018*). At earlier stages of AD, increased population of microglia and astrocytes (microgliosis and astrogliosis) has been observed in diseased regions due to sustained cellular proliferation in response to disturbances, loss of homeostasis, or the accumulation of misfolded proteins (*Kwon and Koh, 2020*; *Keren-Shaul et al., 2017*; *Vandenbark et al., 2021*). Excessive proliferation may lead to the transition of homeostatic microglia to its senescent or disease-associated type, also known as DAM, via the processes mediated by *TREM2-APOE* signaling (*Keren-Shaul et al., 2017*; *Zhou et al., 2020*; *Hu et al., 2021*). Increased number of dystrophic microglia, a form of cellular senescence characterized as beading and fragmentation of the branches of microglia, has been seen in multiple neurodegenerative conditions such as AD, DLB, and TDP-43 encephalopathy (*Streit et al., 2020*). The presence of senescent microglia is believed to ultimately contribute to the failure of brain homeostasis and to clinical symptomatology (*Balusu et al., 2023*; *Hu et al., 2021*; *Lau et al., 2023*).

Oligodendrocytes also associated with spatial tissue vulnerability to all conditions aside ALS (*Figure 3A*). Oligodendrocytes are responsible for the synthesis and maintenance of myelin in the brain (*Armada-Moreira et al., 2015*). Demyelination produces loss of axonal insulation, leading to neuronal dysfunctions (*Armada-Moreira et al., 2015*; *Mot et al., 2018*). Myelin dysfunction may lead to secondary inflammation and subsequent failure of microglia to clear amyloid-β deposition in AD mice models (*Depp et al., 2023*). Oligodendrocytes were shown to be highly genetically associated with PD (*Bryois et al., 2020*; *Feleke et al., 2021*; *Agarwal et al., 2020*). In addition, densities of OPCs showed strong correlations with the atrophy patterns of DLB, EOAD, PS1, and FTLD-TDP type C. OPCs regulate neural activity and harbor immune-related and vascular-related functions (*Akay et al., 2021*). In response to oligodendrocyte damage, OPCs initiate their proliferation and differentiation for the purpose of repairing damaged myelin (*Ohtomo et al., 2018*). In AD, PD, and ALS, the OPCs become unable to differentiate and their numbers decrease, leading to a reduction in myelin production and subsequent neural damage (*Traiffort et al., 2021*; *Spaas et al., 2021*).

We observed (*Figure 3A*) that neuronal abundance distribution is also associated with tissue damage in many neurodegenerative conditions. However, these associations are less strong than for other cell types, except for the ALS case (*Figure 2J*). For this disorder, neuron proportions positively correlated with tissue integrity (i.e., the higher the neuronal proportion, the less atrophy in a region). This observation suggests that increased neuronal presence at brain regions (relative to all considered cell types) may have a protective effect in ALS, making neuronal enriched regions less vulnerable to damage in this disorder. In addition, we observed particularly weak associations between neuronal proportions and tissue damage in all three clinical variants of FTD (bvFTD, nfvPPA, svPPA) and PD (*Figure 3A*), suggesting that these conditions may be primarily associated with supportive cell types (microglia, astrocytes, and oligodendrocytes, respectively; *Figure 2I and K–M*).

## Spatial cell type grouping exposes distinctive disease–disease similarities

Next, we hypothesized that disorders sharing similar biological mechanisms and clinical manifestations present common across-brain patterns of cell type density associations. *Figure 3A* shows a hierarchical taxonomy dendrogram grouping cell types and conditions according to their common brain-wide correlation patterns.

The clustergram analysis revealed distinct grouping patterns among various neurodegenerative conditions. All histologically confirmed dementia conditions formed a separate cluster. Notably, EOAD and mutations in PS1, a prevalent cause of familial EOAD (*Kelleher and Shen, 2017*), grouped together. Interestingly, three clinical subtypes of FTD (bvFTD, nfvPPA, and svPPA) displayed similar patterns of cell type vulnerabilities and diverged into a discrete cluster with PD, separately from ALS and other dementia conditions. However, FTLD-associated pathologies such as TDP-43 proteinopathies

(types A and C), as well as three-repeat and four-repeat tauopathies, showed patterns more similar to those found in DLB and AD-related conditions (EOAD, LOAD, PS1) rather than clinical FTD subtypes from a different dataset. These differences could be attributed to variations in the source dataset; the atrophy maps were derived from different studies and measured by different techniques, which may have introduced discrepancies in results due to different data acquisition tools and protocols (see 'Materials and methods'). Nonetheless, all FTLD-related subtypes and conditions showed the strongest associations with spatial distributions of glial cells, particularly astrocytes and microglia.

Among all cell types, neurons' and OPCs' spatial density distributions were least associated with tissue atrophy in all 13 conditions, subsequently clustering together. Astrocytes and microglia distributions similarly showed the strongest associations with all neurodegenerative conditions (*Figure 3B*), and thus formed a separate cluster while still being related with oligodendrocytes and endothelial cells. Astrocytes and microglia are known to be intimately related in the pathophysiological processes of neurodegenerative disorders (*Kwon and Koh, 2020*). Both are key regulators of inflammatory responses in the central nervous system, and given their role in clearing misfolded proteins, dysfunctions of each of them can result in the accumulation of amyloid-β and tau (*Kwon and Koh, 2020*; *Leyns and Holtzman, 2017*). During the progression of AD and PD, microglia's activation can result in an increased capacity to convert resting astrocytes to reactive astrocytes (*Liddelow et al., 2017*).

Patterns in cellular vulnerability in DLB did not strongly resemble PD without dementia (*Figure 3C*), although both conditions involve alpha-synuclein aggregates (*Kim et al., 2014*). A similar observation can be made for ALS and FTLD. Despite the common presence of TDP-43 abnormal accumulations and their strong genetical overlap (*Ferrari et al., 2011*), ALS did not group together with FTD variants and FTLD-associated pathologies based on patterns of cell–atrophy associations. All these conditions are known to be pathologically linked, often arising from either tau or TDP-43 accumulation; for instance, TDP-43 is the usual cause of svPPA and approximately half of bvFTD cases, while the other half of bvFTD patients and many nfvPPA cases are associated with tau pathology (*Perry and Miller, 2013*). These results emphasize the fundamental role of network topology and other factors beyond the presence of toxic misfolded proteins in developing characteristic tissue loss and cellular vulnerability in neurodegenerative conditions (*Zeighami et al., 2015*; *Iturria-Medina and Evans, 2015*; *Tremblay et al., 2021*; *Zeighami et al., 2019*).

## Discussion

Previous efforts to describe the composition of the brain's different cell populations related to neurodegeneration have been limited to a few isolated regions. In the most systematic study of its kind, here we characterized large-scale spatial associations between canonical cell types and brain tissue loss across cortical and subcortical GM areas in 13 neurodegenerative conditions (including EOAD, LOAD, PD, DLB, ALS, mutations in PS1, and clinical [bvFTD, nfvPPA, svPPA] and pathological [three-repeat and four-repeat tauopathies and TDPP43 proteinopathies types A and C] subtypes of FTLD). Starting from healthy brain levels of gene expression and structural MRI data from the AHBA (*Shen et al., 2012*), and extending our analysis with advanced single-cell-RNA seq-validated cell deconvolution approaches, along with whole-brain atrophy maps from clinically and/or neuropathologically confirmed disorders, we determined that (i) the spatial distributions of non-neuronal cell types, primarily microglia and astrocytes, are strongly associated with the spread tissue damage present in many neurodegenerative conditions; (ii) cells and disorders define major axes that underlie spatial vulnerability, aiding in comprehending heterogeneity behind distinct and similar clinical manifestations/definitions; and (iii) the generated whole-brain maps of cellular abundance can be similarly used for studying the associations between imaging phenotypes and healthy reference cellular levels in other neurological conditions (e.g., neurodevelopmental and neuropsychiatric disorders). Overall, our findings stress the critical need to surpass the current neuro-centric view of brain diseases and the imperative for identifying cell-specific therapeutic targets in neurodegeneration. For further translation and validation, all resulting cell abundance maps and analytic tools are freely shared with the community.

We derived, first to our knowledge, high-resolution maps of cellular abundance/proportion in the adult human healthy brain for six canonical cell types, including astrocytes, neurons, oligodendrocytes, microglia, and endothelial cells. As mentioned, previous cellular analyses of neurological conditions have been restricted to expert-selected isolated brain areas. The invasive nature of expression assays,

requiring direct access to neural tissue, and other numerous scaling limitations have impeded extensive spatial analyses (*Arnatkeviciute et al., 2019*). Earlier studies, also using AHBA data, have shown that spatial patterns in cell type-specific gene expression are associated with both regional vulnerability to neurodegeneration and patterns of atrophy across the brain (*Zeighami et al., 2023*; *Vidal-Pineiro et al., 2020*; *Roshchupkin et al., 2016*; *Zheng et al., 2019*; *Altmann et al., 2020*). Since many neurodegeneration-related genes have similar levels of expression in both affected and unaffected brain areas (*Jackson, 2014*), characterizing changes in tissue loss associated with reference cell type proportions in health may provide a clearer perspective on large-scale spatial patterns of cellular vulnerability. Our maps of cells abundance are available for the scientific and clinical community, potentially allowing researchers to further study spatial variations in cell type density with macroscale phenotypes. These maps can be used in future studies concerning brain structure and function in both health and disease. They can be also explored in the context of other neurological diseases, including neurodevelopmental and psychiatric conditions.

Our results demonstrate that all canonical cell types express vulnerability to dementia-related atrophy of brain tissue, potentially suggesting the disruption of the molecular pathways involving specific cell types can contribute to their observed dysfunctions and subsequent clinical symptomatology (*Balusu et al., 2023*). Previously, transcriptional profiling of prefrontal cortex in AD showed reduced proportions of neurons, astrocytes, oligodendrocytes, and homeostatic microglia (*Lau et al., 2020*). In contrast, bulk-RNA analysis of diseased AD tissues from various human brain regions observed neuronal loss and increased cell abundance of microglia, astrocytes, oligodendrocytes, and endothelial cells (*Johnson et al., 2021*; *Wang et al., 2020*). Furthermore, increased microglial, endothelial cell, and oligodendrocyte population was observed in PD and other Lewy diseases (*Feleke et al., 2021*; *Nido et al., 2020*). Cortical regions exhibiting the most severe atrophy in symptomatic *C9orf72*, *GRN,* and *MAPT* mutation carriers with FTD showed increased gene expression of astrocytes and endothelial cells (*Altmann et al., 2020*). Cortical thinning has been demonstrated to correlate with higher proportions of astrocytes, microglia, oligodendrocytes, OPCs, and endothelial cells in cases of AD compared to controls (*Vidal-Pineiro et al., 2020*; *Kerrebijn et al., 2023*). In line with these results, we observed that regions with increased cell type proportions, particularly for astrocytes and microglia, are strongly associated with GM atrophy in almost all neurodegenerative conditions. This may partly explain the reported cellular proliferation through microglial activation in diseased regions in response to the misfolded protein accumulation or other pathobiological processes (*Keren-Shaul et al., 2017*; *Lau et al., 2023*). As disease progresses, the release of inflammatory agents by sustained microglial activation is believed to be responsible for exacerbating neurodegeneration and clinical symptoms (*Kempuraj et al., 2016*; *Maccioni et al., 2009*). Microglial activation in pair with GM atrophy in frontal cortex was shown to be directly associated with cognitive decline in FTD (*Malpetti et al., 2021*).

Our study has several limitations. Firstly, our analyses were focused on stereotypic atrophy patterns for each disorder. It is known that neurodegenerative diseases are highly heterogeneous, with molecular, phenotypic, and clinical subtypes potentially varying in atrophy patterns (*Fonov et al., 2021*; *Rosenberg-Katz et al., 2013*). Further investigation of cell type signatures across various subtypes not covered in this study and disease stages may better characterize each case. Additionally, comparing our findings with neuropathological assessments of diseased brain tissues in available regions would be beneficial. While the diagnosis of most dementia conditions used in this study has been histologically confirmed, the diagnosis for clinical variants of FTD, ALS, and PD patients was based on clinical and neuroimaging assessments. In addition, it has been observed that cell type-related transcriptional changes are different between sexes (*Mathys et al., 2019*), making future sex-specific analyses indispensable for further understanding of sex-related pathomechanisms. An important consideration is that examined atrophy maps were sourced from different studies (*Supplementary file 2*), with differences in data acquisition protocols (e.g., spatial resolution) and technical procedures (e.g., smoothing level, statistical methods). In complementary analyses, we observed almost identical results after smoothing all disorder-specific images with the same kernel size, while they were already mapped at the same spatial resolution for this study and statistically adjusted by acquisition parameters (e.g., field strength) in original studies. Moreover, cell type deconvolution approaches are varied and limited in their precision (*Dai et al., 2023*). Here, we used a previously validated deconvolution method designed for efficiently estimating cell proportions for six major cell types from bulk mRNA

expression (*McKenzie et al., 2018a*). Conveniently, this method is freely available for researchers (R package, BRETIGEA), which will facilitate reproducibility analyses of our study. Other important considerations are the dynamic nature of gene expression as disease progresses (*Iturria-Medina et al., 2020*; *Hammond et al., 2019*), *postmortem* RNA degradation of the used templates (*Jaffe et al., 2017*), and the subsequent limited ability of bulk RNA sequencing to reflect cell-to-cell variability, which is relevant for understanding cell heterogeneity and the roles of specific cell populations in disease (*Yu et al., 2021*). Lastly, a promising future direction would be to validate our findings with single-cell spatial analyses.

## Materials and methods
### Disorder-specific atrophy maps
Voxel-wise brain atrophy maps in EOAD, LOAD, PD, ALS, DLB, mutation carriers in PS1, clinical variants of FTD, and FTLD pathologies (FTLD-TDP types A and C, three-repeat tauopathy and four-repeat tauopathy) were adopted from open data repositories and/or requested from collaborators (*Harper et al., 2017*; *Dadar et al., 2020*; *Zeighami et al., 2015*; *Dadar and Metz, 2023*), as specified below. Reduction in GM density in diseased atrophy maps relative to controls was measured by VBM and DBM applied to structural T1-weighted MR images, and thus was expressed as *t*-score per voxel (relatively low negative *t*-scores indicate greater GM tissue loss/atrophy) (*Aubert-Broche et al., 2013*; *Ashburner and Friston, 2000*). VBM is a hypothesis-free technique for analyzing neuroimaging data that characterizes regional tissue concentration differences across the whole brain, without the need to predefine regions of interest (*Whitwell, 2009*). DBM is a similar widely used technique to identify structural changes in the brain across participants, which in addition considers anatomical differences such as shape and size of brain structures (*Chung et al., 2001*). See *Supplementary file 2* for study origin, sample size, and imaging technique corresponding to each atrophy map.

MRI data for neuropathological dementias were collected from 186 individuals with a clinical diagnosis of dementia and histopathological (postmortem or biopsy) confirmation of underlying pathology, along with 73 healthy controls (*Harper et al., 2017*). Data were averaged across participants per condition: 107 had a primary AD diagnosis (68 early-onset [<65 y at disease onset], 29 late-onset [≥65 y at disease onset], 10 PS1 mutation carriers), 25 with DLB, 11 with three-repeat-tauopathy, 17 with four-repeat-tauopathy, 12 FTLD-TDP type A, and 14 FTLD-TDP type C (*Harper et al., 2017*). Imaging data were collected from multiple centers on scanners from three different manufacturers (Philips, GE, and Siemens) using a variety of different imaging protocols (*Harper et al., 2017*). Magnetic field strength varied between 1.0 T (*n* = 15 scans), 1.5 T (*n* = 201 scans), and 3 T (*n* = 43 scans) (*Harper et al., 2017*). Pathological examination of brain tissue was conducted between 1997 and 2015 according to the standard histopathological processes and criteria in use at the time of assessment at one of four centers: the Queen Square Brain Bank, London; Kings College Hospital, London; VU Medical Centre, Amsterdam; and Institute for Ageing and Health, Newcastle (*Harper et al., 2017*). Atrophy maps were statistically adjusted for age, sex, total intracranial volume, and MRI strength field and site (*Harper et al., 2017*). Ethical approval for this retrospective study was obtained from the National Research Ethics Service Committee London-Southeast (*Harper et al., 2017*).

MRI data for PD consisting of 3 T high-resolution T1-weighted scans were obtained from the Parkinson's Progression Markers Initiative (PPMI) database (*Parkinson Progression Marker Initiative, 2011*). The PPMI is a multicenter international study with approved protocols by the local institutional review boards at all 24 sites across the United States, Europe, and Australia (*Parkinson Progression Marker Initiative, 2011*). MRI data were acquired in 16 centers participating in the PPMI project using scanners from three different manufacturers (GE medical systems, Siemens, and Philips medical systems). 3 T high-resolution T1-weighted MRI scans from the initial visit and clinical data used in constructing atrophy maps were collected from 232 participants with PD and 118 age-matched controls (*Zeighami et al., 2015*). PD subjects (77 females; age 61.2 ± 9.1) were required to be at least 30 years old or older, untreated with PD medications, diagnosed within the last two years, and to exhibit at least two or more PD-related motor symptoms, such as asymmetrical resting tremor, uneven bradykinesia, or a combination of bradykinesia, resting tremor, and rigidity (*Parkinson Progression Marker Initiative, 2011*). All individuals underwent dopamine transporter (DAT) imaging to confirm a DAT deficit as a

prerequisite for eligibility (*Parkinson Progression Marker Initiative, 2011*). No significant effect of age, gender, or site was found (*Zeighami et al., 2015*).

For ALS, MRI data were collected from 66 patients (24 females; age 57.98 ± 10.84) with both sporadic or familial form of disease from centers of the Canadian ALS Neuroimaging Consortium ( ClinicalTrials.gov NCT02405182), which included 3 T MRI sites in University of Alberta, University of Calgary, University of Toronto, and McGill University (*Dadar et al., 2020*; *Kalra et al., 2020*). Patients were included if they were diagnosed with sporadic or familial ALS, and meet the revised El Escorial research criteria (*Brooks et al., 2000*) for possible, laboratory-supported, or definite ALS (*Kalra et al., 2020*). Patients underwent a neurological exam administered by a trained neurologist at each partic-ipating site (*Kalra et al., 2020*). All participants gave written informed consent, and the study was approved by the health research ethics boards at each of the participating sites (*Dadar et al., 2020*). Participants were excluded if they had a history of other neurological or psychiatric disorders, prior brain injury, or respiratory impairment resulting in an inability to tolerate the MRI protocol (*Dadar et al., 2020*). Participants with primary lateral sclerosis, progressive muscular atrophy, or FTD were also excluded from the study (*Kalra et al., 2020*). Normative aging as well as sex differences were regressed out from data prior the map construction (*Dadar et al., 2020*).

For clinical subtypes of FTD, atrophy maps were obtained from the open-access database (*Dadar and Metz, 2023*). These maps were derived from MRI data from the Frontotemporal Lobar Degen-eration Neuroimaging Initiative (FTLDNI AG032306; part of the ALLFTD). As described in separate studies (*Dadar et al., 2021*; *Dadar et al., 2022*), the data used for constructing these atrophy maps consisted of 136 patients diagnosed with FTD, alongside 133 age-matched control partici-pants. Participants were previously stratified into groups according to their clinical variant of FTD: 70 patients were diagnosed with the behavioral variant, 36 with the semantic primary progressive aphasia, and 30 with the non-fluent primary progressive aphasia (*Staffaroni et al., 2019*; *Dadar et al., 2021*). 3 T structural images were collected on the following three sites: University of California San Francisco, Mayo Clinic, and Massachusetts General Hospital (*Staffaroni et al., 2019*). Patients were referred by physicians or self-referred, and all underwent neurological, neuropsychological, and functional assessment with informant interview (*Staffaroni et al., 2019*). All individuals received their diagnoses during a multidisciplinary consensus conference using established criteria: Neary criteria (*Neary et al., 1998*) or, depending on the year of enrollment, the recently published consensus criteria for bvFTD (*Rascovsky et al., 2011*) and PPA (*Gorno-Tempini et al., 2011*). Histological analysis was conducted to assess whether patients might have AD pathology since both conditions present the overlap of clinical symptoms (*Staffaroni et al., 2019*). All subjects provided informed consent, and the protocol was approved by the institutional review board at all sites (*Staffaroni et al., 2019*).

## Mapping gene expression data

To construct a comprehensive transcriptome atlas, we used mRNA microarray gene expression data from the AHBA (*Shen et al., 2012*). The AHBA included anatomical and histological data collected from six healthy human specimens with no known neurological disease history (one female; age range 24–57 y; mean age 42.5 ± 13.38 y) (*Shen et al., 2012*). Two specimens contained data from the entire brain, whereas the remaining four included data from the left hemisphere only, with 3702 spatially distinct samples in total (*Shen et al., 2012*). The samples were distributed across cortical, subcortical, brainstem, and cerebellar regions in each brain, and the expression levels of more than 20,000 genes were quantified (*Shen et al., 2012*). mRNA data for specific brain locations were accompanied by structural MR data from each individual and were labeled with Talairach native coordinates (*Talairach and Szikla, 1980*) and MNI coordinates (*Evans et al., 1994*), which allowed us to match samples to imaging data.

Following the validated approach in *Gryglewski et al., 2018*, missing data points between samples for each MNI coordinate were interpolated using Gaussian process regression, a widely used method for data interpolation in geostatistics. The regression is performed as a weighted linear combination of missing mRNA, with the weights decreasing from proximal to distal regions. MNI coordinates for predicting mRNA values were taken from the GM regions of the AAL atlas. Spatial covariance between coordinates from the available 3072 AHBA tissue samples and coordinates from the AAL atlas was estimated via the quadratic exponential kernel function. mRNA expression at each MNI coordinate

was then predicted by multiplying AHBA gene express values that corresponded to specific probes to kernel covariance matrix divided by the sum of kernels.

## Cell type proportion estimation

Densities for multiple canonical cell types were estimated at the GM by applying an R-package BRETIGEA, with known genetic markers to the transcriptome atlas (*McKenzie et al., 2018a*). This eigengene decomposition-based deconvolution method was designed for estimating cell proportions in bulk gene expression data for six major cell types: neurons, astrocytes, oligodendrocytes, microglia, endothelial cells, and OPCs (*McKenzie et al., 2018a*; *Chikina et al., 2015*). We chose 15 representative gene markers per each cell type (90 in total) from the BRETIGEA human brain marker gene set and then selected those genes that were also present in the AHBA gene expression database with matching gene probes. This resulted in 80 cell type-related gene markers that were used in missing data interpolation and the deconvolution proportion estimation analysis (*Supplementary file 3*). For each voxel, each cell type proportion value was normalized relative to the sum of all six cell types and the sum was scaled relative to the GM density. We then registered data into MNI and volumetric space using the ICBM152 template (*Evans et al., 1994*).

For the correlation analysis, cell densities were averaged over 118 anatomical regions in GM defined by the extended AAL atlas (*Supplementary file 1*; *Tzourio-Mazoyer et al., 2002*). We repeated the correlation analysis for the 98 regions from the DKT atlas (*Figure 3—figure supplement 1*; *Desikan et al., 2006*).

## Data analysis

We constructed a $6 \times 13$ correlation matrix by computing inter-regional Spearman's correlations between spatial distributions of the 6 canonical cell types and patterns of atrophy in 13 neurodegenerative conditions. Correction for multiple comparisons using the FDR was conducted using the Benjamini–Hochberg method, with a significance threshold of 0.05. Shapiro–Wilk tests were used to examine the normality of data distribution. Hierarchical clustering analyses were applied using in-built MATLAB function for data visualization. Cells and conditions were clustered together based on estimated averaged linkage Euclidian distance between their correlation values.

## Acknowledgements

This project was undertaken thanks in part to the following funding awards to YIM: the Canada Research Chair tier 2, the CIHR Project Grant 2020, the Weston Family Foundation's Transformational Research in AD 2020, and the New Investigator start-up grant from McGill University's Healthy Brains for Healthy Lives Initiative (HBHL, Canada First Research Excellence Fund). VP was supported by the Laszlo & Etelka Kollar Fellowship from the Faculty of Medicine and Health Sciences at McGill University and partly supported by the HBHL's Theme 1 Discovery fund 2022–2025. In addition, we used the computational infrastructure of the McConnell Brain Imaging Center at the Montreal Neurological Institute, supported in part by the Brain Canada Foundation, through the Canada Brain Research Fund, with the financial support of Health Canada and sponsors.

## Additional information

### Funding

| Funder | Grant reference number | Author |
| --- | --- | --- |
| Faculty of Medicine, McGill University | Laszlo & Etelka Kollar Fellowship | Veronika Pak |
| Canada First Research Excellence Fund | HBHL's Theme 1 Discovery fund 2022-2025 | Yasser Iturria-Medina |
| Canada Research Chairs | Tier 2 | Yasser Iturria-Medina |
| Weston Family Foundation | Transformational Research in AD 2020 | Yasser Iturria-Medina |

| Funder | Grant reference number | Author |
|---|---|---|
| Canada First Research Excellence Fund | HBHL's New Recruit Start-Up Supplements | Yasser Iturria-Medina |
| Canadian Institutes of Health Research | CIHR Project Grant 2020 | Yasser Iturria-Medina |

The funders had no role in study design, data collection and interpretation, or the decision to submit the work for publication.

## Author contributions

Veronika Pak, Conceptualization, Data curation, Formal analysis, Investigation, Visualization, Methodology, Writing - original draft, Writing - review and editing; Quadri Adewale, Conceptualization, Data curation, Formal analysis, Methodology; Danilo Bzdok, Writing - review and editing; Mahsa Dadar, Yashar Zeighami, Resources, Writing - review and editing; Yasser Iturria-Medina, Conceptualization, Supervision, Funding acquisition, Methodology, Writing - original draft, Project administration, Writing - review and editing

## Author ORCIDs

Veronika Pak http://orcid.org/0009-0008-6305-2541
Quadri Adewale http://orcid.org/0000-0001-5090-6140
Mahsa Dadar http://orcid.org/0000-0003-4008-2672
Yasser Iturria-Medina http://orcid.org/0000-0002-9345-0347

## Ethics

Our study used human data previously preprocessed for other studies. Informed consent and ethics approval obtained for those studies are described in the Materials and Methods section.

Reviewer #1 (Public Review): https://doi.org/10.7554/eLife.89368.3.sa1
Reviewer #3 (Public Review): https://doi.org/10.7554/eLife.89368.3.sa2
Author Response https://doi.org/10.7554/eLife.89368.3.sa3

# Additional files

## Supplementary files

- Supplementary file 1. Cortical and subcortical regions from the AAL atlas.
- Supplementary file 2. Origin of each disorder-associated *t*-statistic map.
- Supplementary file 3. Eighty cell type-related gene markers provided by the BRETIGEA R package.
- MDAR checklist

## Data availability

All data needed to evaluate the conclusions in the paper are present in the article and/or the supplementary materials. The BRETIGEA R package is available at *McKenzie et al., 2018a*; *McKenzie et al., 2018b*. The Allen Human Brain Atlas data is available at https://human.brain-map.org/static/download. Atrophy maps for pathologically confirmed dementia are available at NeuroVault. Raw demographic and MRI data from PD and ALS patients can be accessed at https://www.ppmi-info.org/ and http://calsnic.org/ (ClinicalTrials.gov Identifier: NCT02405182), respectively. Atrophy maps for clinical variants of FTD are available at Zenodo. Raw data from the FTLDNI initiative can be downloaded from the Laboratory of Neuroimaging (LONI) Image Data Archive. The cells abundance maps from this study are freely shared with the community and can be found at our lab's GitHub space (copy archived at *neuropm-lab, 2023*).

The following previously published datasets were used:

| Author(s) | Year | Dataset title | Dataset URL | Database and Identifier |
|-----------|------|---------------|-------------|-------------------------|
| Dadar M, Metz A | 2023 | Atrophy Pattern Maps of Frontotemporal Dementia variants (bvFTD, svPPA, pnfaPPA) | https://doi.org/10.5281/zenodo.10383492 | Zenodo, 10.5281/zenodo.10383492 |
| Harper L, Bouwman F, Burton EJ, Barkhof F, Scheltens P, O'Brien JT, Fox NC, Ridgway GR, Schott JR | 2017 | Patterns of atrophy in pathologically confirmed dementias: a voxelwise analysis | https://identifiers.org/neurovault.collection:1818 | NeuroVault, neurovault.collection:1818 |

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
