## [Editor Report · eLife assessment]

Pak et al. examined the relationship between the most common spatial patterns of neurodegeneration and transcriptional markers of the density of different cell types in the cerebral cortex. This **valuable** study uses innovative methods to provide **convincing** evidence that patterns of gray matter loss in various forms of dementia are correlated with the anatomical distribution of non-neuronal cell types.

---

## [Referee Report · Reviewer #1 (Public Review)]

This study explores the relationship between neurodegeneration's most common spatial patterns and the density of different cell types in the cerebral cortex. The authors present data showing that atrophy patterns in Alzheimer's disease and Frontotemporal dementia (FTD) strongly associated with the abundance of astrocytes and microglia. This work (the original manuscript and the revision) takes a step in the right direction by emphasizing the critical role that cells other than neurons play in the degeneration patterns observable with neuroimaging.

Comments on revised version:

I appreciate the revisions the authors made to address my main comments:

- adding whole-brain maps showing cellular abundance and atrophy

- stratifying the FTD group into the three clinically defined categories bvFTD (behavior-variant), nfvPPA (nonfluent/agrammatic-variant primary progressive aphasia), and svPPA (semantic-variant primary progressive aphasia).

I reiterate my agreement with the authors that this work demonstrates the need to "surpass the current neuro-centric view of brain diseases and the imperative for identifying cell-specific therapeutic targets in neurodegeneration".

---

## [Referee Report · Reviewer #3 (Public Review)]

This study is a fine example of a recent productive trend in the integration of neuroimaging and molecular biology of the brain: in brief, overlaying some neuroimaging data (usually from a large cohort) onto the high spatial resolution gene expression in the Allen Human Brain Atlas data, derived from 6 individuals. By projecting structural MRI images over cell type proportions identified in the Allen data, the authors can represent various diseases in terms of their spatially-associated cell types. The result has implications for prioritizing the contributions of various cell types to each disease and creates an even-handed cell type profile through which the 11 diseases can be compared.

---

## [Author Response]

The following is the authors’ response to the original reviews.

Reviewer 1: I would have preferred to see more figures with brain images showing the cellular abundance maps and the atrophy maps. Without being able to see these figures, it's difficult for the reader to (1) validate the atrophy patterns or (2) gain intuition about how the cellular abundance maps vary across the brain. The images in Figure 1C give a small preview, but I'd like to see these maps in their entirety on the brain surface or axial image slices.

(1) We added brain surface visualization plots of the voxel-wise cellular abundance maps to Figure 1 (lateral, dorsal, and ventral views of both hemispheres). To illustrate how their spatial distributions are associated with brain tissue damage, in Figure 2, we have also added brain surface visualizations of regional values from the atrophy t-statistic maps for the thirteen neurodegenerative conditions and the cell-type map most strongly associated with each condition. These plots allow us to observe variability across the cell-type density and atrophy maps, as well as to visually validate and compare how the patterns vary across the brain.

Reviewer 1: FTD is an umbrella category for a family of distinct clinical syndromes with different atrophy patterns. It doesn't seem a good idea to take the average of all subjects in this group to form a single atrophy map. Instead, different average maps for each syndrome should be provided.

(2) Considering the heterogeneity of clinical FTD syndromes, we addressed the reviewers' concerns about using the averaged atrophy map across all patients with an FTD diagnosis. As suggested, we accessed different atrophy maps for each major variant of clinical FTD, including behavioral FTD (n = 70), as well as the semantic (n = 36) and nonfluent variants of primary progressive aphasia (n = 30). These maps are based on data from the participants from the same dataset of the Frontotemporal Lobar Degeneration Neuroimaging Initiative (FTLDNI) that we originally used. Similar to our previous results using the atrophy map averaged over all FTD patients, the analysis showed significant associations of atrophy patterns with cell type densities in all three major variants (see Figure 3A). Notably, these new findings offer insights into specific differences in spatial vulnerability of different cell-types across the variants of FTD, each characterized by unique symptoms, clinical manifestations, and atrophy patterns. In response to these additions, we have updated all figures, results, and interpretations accordingly.

Reviewer 2: In the abstract, the list of neurodegenerative disorders should be edited: frontotemporal dementia is an umbrella clinical syndrome, not a neurodegenerative disorder. Frontotemporal lobar degeneration (FTLD) is a neurodegenerative disorder, and many tauopathies are FTLDs. While the authors grab their definitional classes from various sources (i.e., published cohort, and other studies), the reader fatigues to understand the population that is being assessed.

(3) To address potential confusion arising from the inclusion of atrophy maps from FTLD patients across two different studies, stratified based on both clinical and pathological criteria, we added clarifications regarding the assessed population and the used definitions. We used the term FTD when addressing the clinical syndromes, and the term FTLD was employed when referencing the histologically confirmed neurodegenerative pathologies. In addition, we added details on the diagnostic criteria employed for participant recruitment in the FTLDNI cohort, which data we used for atrophy maps in clinical subtypes of FTD. Lastly, throughout the text and within the figures, we systematically refined the nomenclature for FTLD pathological types, categorizing them based on their known definitions used in literature and type of proteinaceous inclusions (FTLD- 3-repeat and 4-repeat tauopathies and FTLD-TDP types A and C).

Reviewer 1: The results section contains perhaps too much interpretation. While the information that's provided serves as an interesting review (e.g., the discussion of the blood-brain barrier), the discussion may be a better place for this.

(4) We removed sentences with excessive interpretation but insisted on including those outlining the fundamental functions of cell types and their literature-based relevance to neurodegenerative diseases in the Results section, clarifying the significance of our findings to the readers.

Reviewer 2: The authors based their methodology on the use of a deconvolutional cell classifier; however, do not extensively recognize that their data on gene expression are based on normal brain levels rather than on diseased ones.

(5) We acknowledged that the gene expression data is based on normal human brain levels in figure titles and all sections of the paper (Introduction, Results, Discussion, Methods) to remind the readers that the analysis shows how changes in gray matter tissue in diseased brains correlates with healthy reference levels of cellular density.

Reviewer 2: More information in the text needs to be provided regarding the method used to infer gene expression levels at non-sampled brain locations. The reader should not be forced to read reference 40 or investigate the methods section. Figure 1 schematics do not sufficiently explain the used method.

(6) We added clarifications/references about the used Gaussian progress regression for imputing gene expression (Results and figure titles).

Reviewer 2: Also, while predicted levels are uniquely based on patterns of brain atrophy, it is not possible to know whether this strategy is generalizable to all diseases (for instance, it is known that pure DLB, PD and ALS are not associated with extensive brain atrophy), or even adequately comparable between subtypes of diseases within the same class (e.g., different forms of FTLD). The authors do not acknowledge that only data based on true neuropathological assessment may prove whether their findings are true.

(7) Although diagnoses of most dementia conditions used in our study were histologically confirmed, we added acknowledgement about the importance of neuropathological assessment (Discussion section).